# Diagnostic Dilemma of Biliopancreatic Contrast-Enhanced Harmonic Endoscopic Ultrasonography

**DOI:** 10.3390/diagnostics12081983

**Published:** 2022-08-16

**Authors:** Keisuke Kanazawa, Masafumi Chiba, Masayuki Kato, Yuji Kinoshita, Takafumi Akasu, Hiroaki Matsui, Nana Shimamoto, Youichi Tomita, Takahiro Abe, Shintaro Tsukinaga, Masanori Nakano, Yuichi Torisu, Hirobumi Toyoizumi, Kazuki Sumiyama

**Affiliations:** 1Department of Endoscopy, The Jikei University School of Medicine, Tokyo 105-8461, Japan; 2Division of Gastroenterology and Hepatology, Department of Internal Medicine, The Jikei University School of Medicine, Tokyo 105-8461, Japan

**Keywords:** ultrasound contrast agents, endosonography, diagnostic performance, biliary tract diseases, pancreatic diseases

## Abstract

Background: The utility of contrast-enhanced harmonic endoscopic ultrasonography (CH-EUS) alone in the biliopancreatic region appears to be limited because it is highly dependent on the experience and skill of the endoscopist. Therefore, the present study aimed to validate the efficacy of CH-EUS in clinical practice. Methods: Between January 2018 and March 2019, 301 consecutive patients who underwent CH-EUS were prospectively enrolled in this study. The diagnostic performance of CH-EUS was compared with that of dynamic computed tomography (CT), magnetic resonance imaging (MRI), and all combinations (i.e., CH-EUS, dynamic CT, and MRI) using a Bonferroni correction. A multiple logistic regression analysis was performed to extract each disease that allowed the CH-EUS diagnosis to be consistent with the final diagnosis. Results: In multiple comparisons of diagnostic performance, no significant differences were observed among dynamic CT, MRI, and CH-EUS (*p* = 1.00), but the diagnostic performance was significantly higher when all modalities were combined (*p* < 0.001). Moreover, only intraductal papillary mucinous neoplasm comprising adenoma or carcinoma (IPMN, n = 161) showed significance with respect to the agreement with the final diagnosis (*p* = 0.006). Conclusions: Our results showed that CH-EUS-based diagnosis of IPMN may be possible in clinical practice. On the contrary, to accurately diagnose biliopancreatic diseases other than IPMN, comprehensive diagnosis using multiple modalities may be necessary, rather than relying on CH-EUS alone.

## 1. Introduction

Many studies have reported that contrast-enhanced harmonic endoscopic ultrasonography (CH-EUS) is useful in the biliopancreatic region, and today, it is widely considered an indispensable test [1,2,3,4]. However, the utility of CH-EUS in the diagnosis of biliopancreatic diseases appears to be limited because it depends on the skill and experience of the endoscopist; hence, diagnostic results may be divergent [4,5,6,7]. Therefore, although a shift toward a more objective quantitative diagnosis has occurred, CH-EUS still appears to be a difficult procedure for the average endoscopist, with the exception of a few specialists, to perform in clinical practice [6,8,9].

Although endoscopic ultrasound-guided fine needle aspiration (EUS-FNA) is becoming an increasingly typical tool for the pathological diagnosis of biliopancreatic diseases [3,10,11], CH-EUS plays a significant adjunctive role in cases of false-negative EUS-FNA where a puncture is created to avoid vessels and cystic lesions [12,13,14].

While the noninvasive but limited and ancillary roles of CH-EUS have been highlighted, the biliopancreatic diseases that truly favor CH-EUS diagnosis and their influencing factors remain unknown in clinical practice owing to the fact that most previous reports have compared or evaluated the diagnostic performance of CH-EUS within a single disease group (e.g., comparison of suspected pancreatic cancer only) [1,2,4].

Thus, the present study aimed to validate the efficacy of CH-EUS in actual clinical practice.

## 2. Materials and Methods

### 2.1. Study Design and Patients

The present prospective single-center study included consecutive patients with suspected biliopancreatic disease who underwent CH-EUS between January 2018 and March 2019. A detailed breakdown of each disease is provided in Table 1. Most patients who visit our hospital are referred from neighboring hospitals and are initially assigned to a department according to the suspected disease. Hence, the number of biliopancreatic EUS procedures performed at our hospital is approximately 500–800 annually. Thus, when the primary physician in the Department of Gastroenterology or Gastroenterological Surgery suspects biliopancreatic disease, primary physicians, at their discretion, request the endoscopy department to perform a CH-EUS. Furthermore, the endoscopist who performs EUS is usually different from the primary physicians of the respective departments who request EUS. In the present study, the inclusion criteria were all cases wherein each primary physician determined that CH-EUS was necessary. Exclusion criteria were patients with an egg allergy, critical cardiopulmonary disease, or a cardiopulmonary arteriovenous shunt. Written informed consent to undergo CH-EUS was obtained from all the patients. Similarly, at the discretion of each primary physician, dynamic computed tomography (CT) and magnetic resonance imaging (MRI) were also performed in parallel, if necessary (Table 1).

This study was approved by the Human Subjects Committee of Jikei University School of Medicine (ID no. 29–192 (8808)) and was subsequently registered with the University Hospital Medical Information Network Clinical Trials Registry (identification no. UMIN 000030735). This study was conducted according to the ethical principles of the Declaration of Helsinki (Fortaleza revision) and in accordance with strengthening the reporting of observational studies in epidemiology (STROBE) [15].

### 2.2. Ultrasound Contrast Agent and CH-EUS Procedures

The ultrasound contrast agent used in this study was Sonazoid (Daiichi-Sankyo, Tokyo, Japan; GE Healthcare, Tokyo, Japan). The contrast agent was reconstituted in 2 mL of sterile water, after which a dose of 0.015 mL/kg bodyweight was injected intravenously, followed by flushing with 5 mL of heparinized saline solution. The average examination time was approximately 2 min after contrast agent injection.

CH-EUS was performed by 4 experts with >10 years of EUS experience and by 10 trainees with <10 years of EUS experience. All trainees performed EUS under the direct supervision of experts (Table 1), and all the CH-EUS diagnoses were performed by experts with a board certification in Japan Gastroenterological Endoscopy Society. All cases were examined with a UCT-260 convex scope and an EU-ME2 EUS processor in harmonic detection mode (Olympus Medical Systems, Tokyo, Japan). The transmit frequency was set to 4.7 MHz, and the mechanical index value was set at 0.25–0.30 [16]. All patients who underwent EUS were conscious but sedated with intravenous midazolam and pethidine during the EUS procedure.

### 2.3. Determination of Diagnosis by Dynamic CT, MRI, and CH-EUS

CT and MRI data were assessed independently by board-certified radiologists. All the diagnoses of CH-EUS, even if a trainee performed the CH-EUS, were performed by experts with a board certification in Japan Gastroenterological Endoscopy Society.

### 2.4. Basis for Diagnosis of Each Disease by CH-EUS

For a CH-EUS-based diagnosis of each disease, four contrast patterns were specifically observed: hypoenhancement, isoenhancement, hyperenhancement, and no enhancement. Regarding intraductal papillary mucinous neoplasms (IPMN), intraductal papillary mucinous carcinoma was defined as follows: the presence of structures larger than 5 mm that were contrasted in the lumen of the cyst or in the lumen of the main pancreatic duct after contrast agent administration. Intraductal papillary mucinous adenoma was defined as follows: the absence of structures in the lumen of the cyst or in the lumen of the pancreatic duct, or the presence of structures that were not contrasted [12,17,18]. Pancreatic cancer was defined as a lesion that is hypodense compared with the adjacent pancreatic parenchyma [14]. Pancreatic neuroendocrine neoplasm was defined as the presence of a hypoechoic tumor in the pancreas that exhibited homogeneous early enhancement [2,19]. Gallbladder carcinoma was defined as an intraluminal mass in the gallbladder that exhibited irregular intratumoral vessel enhancement or a perfusion defect [4].

On the contrary, if a lesion did not fit the above contrast patterns, another diagnosis was performed.

### 2.5. Gold Standard for Final Diagnosis

The final diagnosis was based on surgical pathology, pathology of EUS-FNA, biopsy from a metastatic or direct lesion, or an overall determination established by the clinical course with an observation period of >6 months.

### 2.6. Outcome Measures

The primary endpoint of the present study was to identify factors that determine whether CH-EUS is useful in clinical practice. The secondary endpoints were: (1) comparison of diagnostic performance among CH-EUS, dynamic CT, and MRI in biliopancreatic diseases in the same patients; and (2) identification of the best diagnostic method, other than invasive EUS-FNA, which would result in a diagnosis that would be most likely to match the final diagnosis.

### 2.7. Statistical Analysis

To assess the diagnostic performance of dynamic CT, magnetic resonance cholangiopancreatography (MRCP), CH-EUS, and all combinations (i.e., CH-EUS, dynamic CT, and MRI). The area under the receiver operating characteristic (AUC) curves was calculated. One-way analysis of variance with Bonferroni correction was also used for multiple comparisons of AUC values. To extract factors that determine whether CH-EUS is useful in clinical practice, multiple logistic regression analysis was performed. The correspondence between the final diagnosis and the CH-EUS-based diagnosis was defined as the dependent variable, whereas “age”, “sex,” and “disease type” were the independent variables. Missing values were excluded for a complete case analysis. Regarding the sample size calculation, assuming sensitivity of 70% in the CH-EUS group and 80% in all combination groups, with a type I error of 0.05 (two-sided) and a power of 0.8, a minimum of 294 patients in each group was required. All analyses were performed using Stata version 15 (StataCorp LP; College Station, TX, USA), and two-sided *p* values < 0.05 indicate statistical significance.

## 3. Results

### 3.1. Patients and CH-EUS Procedures

During the study period, 301 consecutive cases that underwent CH-EUS were enrolled in this study (Table 1). A detailed breakdown of each disease is provided in Table 1. Finally, of the patients who requested CH-EUS, none were excluded from this study. The proportion of pathologic diagnoses as the final diagnosis was 36.2%. Of the 301 patients who underwent CH-EUS, the number of diagnoses by dynamic CT and MRI was 197 (65.5%) and 265 (88.0%), respectively. The endoscopists who participated included 10 trainees and 4 experts (Table 1). Adverse events associated with the EUS procedure alone occurred in 1.7% of patients, whereas side effects of the contrast agent were not observed (Table 1).

### 3.2. Diagnostic Performance of Dynamic CT, MRI, and CH-EUS for Determining Benign and Malignant Tumors

Of the 301 patients, 197 underwent both DCT and CH-EUS, 265 underwent both MRI and CH-EUS, and 161 underwent MRI, DCT, and CH-EUS. Of the 301 cases that underwent CH-EUS, the diagnostic performance of dynamic CT, MRCP, CH-EUS, and all combinations (i.e., CH-EUS, dynamic CT, and MRI) for the detection of malignancy showed sensitivities of 71.3%, 70.0%, 70.4%, and 80.6% and specificities of 67.0%, 73.1%, 78.8%, and 90.7%, respectively. The positive predictive values (PPVs) for dynamic CT, MRCP, CH-EUS, and all combinations were 66.3%, 57.3%, 65.0%, and 82.9%, respectively, while the negative predictive values (NPVs) were 71.9%, 82.6%, 82.6%, and 89.3%, respectively (Table 2). All the combinations exhibited sensitivity, specificity, PPV, and NPV of >80% (Table 2).

The AUC values of dynamic CT, MRCP, CH-EUS, and all combinations were 0.69, 0.68, 0.70, and 0.81, respectively (Figure 1). Multiple comparisons of the AUC values revealed no significant differences among dynamic CT, MRI, and CH-EUS (*p* = 1.00), and the AUC was significantly higher only for all combinations (*p* < 0.001).

Of the 301 patients, 197 underwent both dynamic CT and CH-EUS, 265 underwent both MRI and CH-EUS, and 161 underwent MRI, dynamic CT, and CH-EUS.

AUC, area under the curve; CT, computed tomography; MRI, magnetic resonance imaging; CH-EUS, contrast-enhanced harmonic endoscopic ultrasonography.

### 3.3. Typical and Atypical Contrast-Enhanced Harmonic Endoscopic Ultrasonography Imaging

Intraductal papillary mucinous adenoma and carcinoma showed higher percentages of typical contrast at 85.3% and 87.5%, respectively. In contrast, pancreatic cancer, pancreatic neuroendocrine neoplasm, and gallbladder carcinoma showed lower percentages of typical contrast at 67.7%, 75.0%, and 68.8%, respectively (Table 3 and Figure 2).

### 3.4. Extraction of Factors That Determine the Utility of CH-EUS

After multiple logistic regression analysis of the correspondence between the final and CH-EUS-based diagnoses as the outcome, no significant differences in age, sex, and disease type other than IPMN comprising adenoma (IPMA) or carcinoma (IPMC) were observed. In contrast, only IPMN showed significance for agreement with the final diagnosis (odds ratio, 6.91; 95% confidence interval, 1.76–27.12; *p* = 0.006) (Table 4).

## 4. Discussion

In the present prospective study of consecutive cases, IPMN consisting of adenoma (IPMA) or carcinoma (IPMC) was the only significant factor that determined whether the CH-EUS-based diagnosis matched the final diagnosis. In contrast, no significant differences were observed among dynamic CT, MRI, and CH-EUS, and the diagnostic correspondence was significantly higher only for all combinations.

Regarding diagnostic performance, it may be difficult for CH-EUS alone to contribute to the final diagnosis of various types of biliopancreatic diseases in clinical practice because the diagnostic performance of CH-EUS did not differ from that of other modalities. Moreover, despite compliance with the typical enhancement pattern of each disease, the diagnostic performance of CH-EUS alone was unsatisfactory because atypical enhancement patterns occurred in approximately 30% of the cases, with the exception of IPMN. In practice, the reasons for the overlap of atypical contrast patterns (e.g., hypoenhancement of pancreatic neuroendocrine tumors) are unknown, and it may be difficult for general endoscopists, other than some specialists, to comprehensively judge these patterns [2,4,5,6,19]. CH-EUS is a noninvasive and attractive pretreatment diagnostic tool; however, it appears to be limited in its diagnostic ability due to its lack of universal objectivity because of atypical contrast patterns [5,7,9]. Thus, a comprehensive diagnosis in combination with other modalities is required in clinical practice, thereby providing awareness regarding atypical contrast patterns. Moreover, pathological diagnosis using the more invasive EUS-FNA or endoscopic retrograde cholangiopancreatography may be strongly considered as a next step if the malignant biliopancreatic disease is suspected in the presence of atypical contrast patterns [10,11,21].

Although EUS-FNA is the mainstay of pretreatment diagnosis of biliopancreatic diseases [10,11], the uncertainty in diagnostic accuracy for small masses (<10 mm) and the degree of invasiveness in patients are not negligible in some cases (e.g., peritoneal seeding of cystic lesions and risk of peritonitis) [12,22]. While diagnostic imaging is important in such situations, our results suggest that all combinations of diagnostic imaging modalities, rather than a single modality, may be useful for general endoscopists to diagnose biliopancreatic diseases in clinical practice.

Based on the results of the multiple logistic regression analysis, IPMN was the only significant factor for CH-EUS-based diagnosis to match the final diagnosis. The results suggest that the diagnosis of IPMN by CH-EUS allows for an easy and objective determination of malignancy by general endoscopists because in the consecutive cases in this study, the typical contrast patterns of IPMA and IPMC accounted for more than 85% of the cases [1,8,23,24,25,26,27,28]. In contrast, diseases other than IPMN may be diagnosed using CH-EUS only by experts with advanced knowledge and skills owing to the atypical contrast patterns being approximately 30% higher [2,5,9].

The present study has several limitations. First, since this study was performed at a single center, multicenter prospective studies are needed for external validation. Second, we did not assess the differences in competency among individual endoscopists (i.e., knowledge and skills). Third, although the missing values arising from the judgment of the requesting primary physician other than the endoscopist had occurred at random in dynamic CT and/or MRI examination, this number of missing values reduced estimation accuracy and statistical power. Finally, the subjectivity of each endoscopist may have led to CH-EUS diagnostic errors, since no objective quantitative analysis (e.g., time intensity curve analysis) was performed for diagnosis by CH-EUS.

## 5. Conclusions

In conclusion, the present study showed that CH-EUS is promising for the accurate diagnosis of IPMN. However, in clinical practice, to diagnose biliopancreatic diseases other than IPMN, a comprehensive determination based on a combination of multiple modalities is necessary, rather than relying on CH-EUS alone.

## Figures and Tables

**Figure 1 diagnostics-12-01983-f001:**
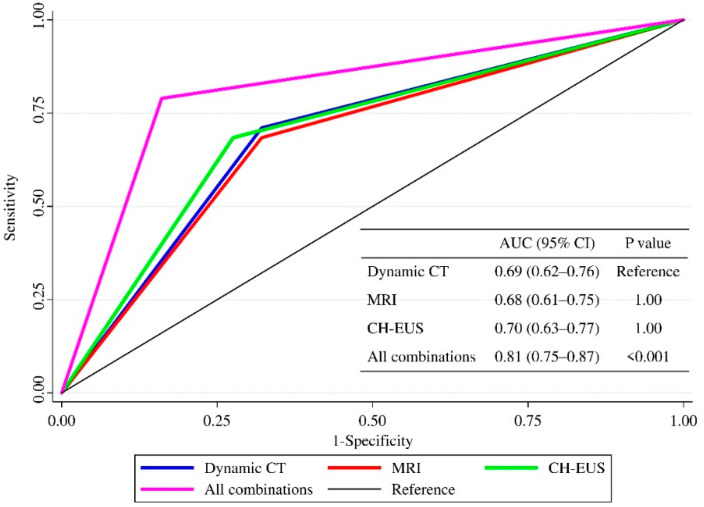
Comparison of the diagnostic performance among dynamic CT, MRCP, CH-EUS, and all combinations (i.e., CH-EUS, dynamic CT, and MRI) using Bonferroni correction (n = 301) during complete case analysis.

**Figure 2 diagnostics-12-01983-f002:**
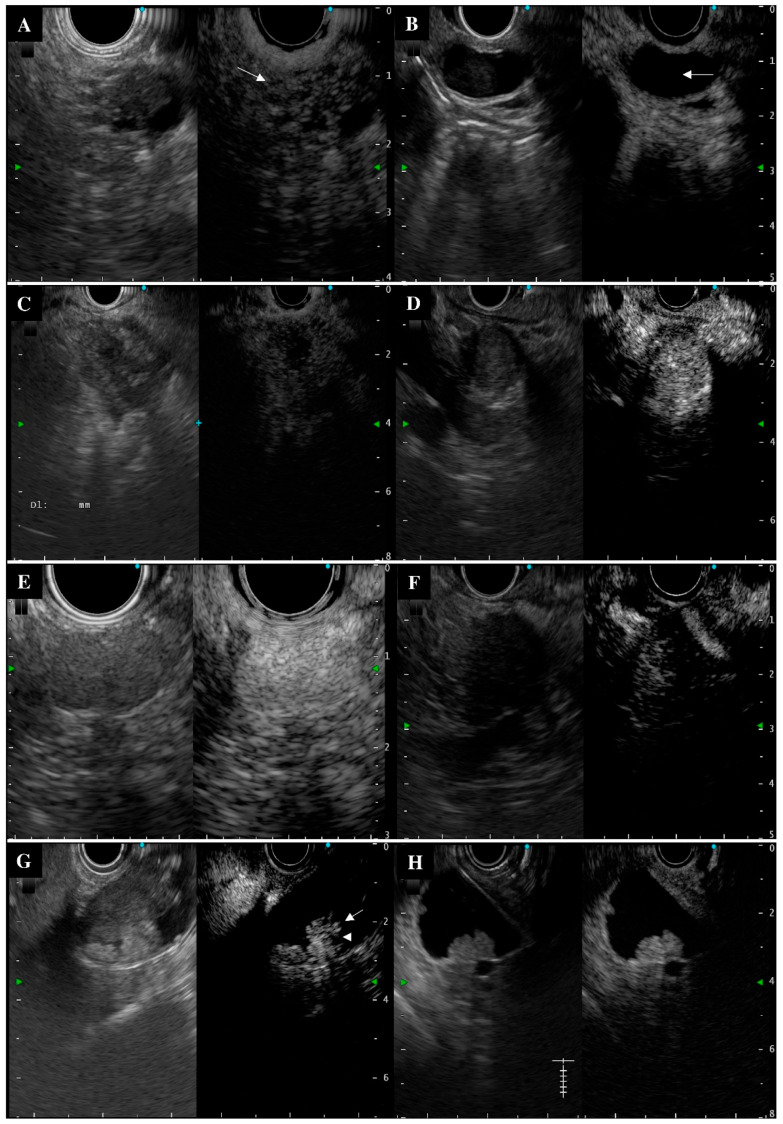
Typical and atypical contrast-enhanced harmonic endoscopic ultrasonography (CH-EUS) imaging of the biliopancreatic lesions (Left: fundamental B mode as a monitor image; Right: CH-EUS mode). (**A**) Typical intraductal papillary mucinous carcinoma imaging with hyperenhancement and a heterogenous pattern of the mural nodule (white arrow, 120 s after contrast infusion). Pathologic diagnosis by surgery: intraductal papillary mucinous carcinoma, invasive. (**B**) Typical intraductal papillary mucinous adenoma imaging with an avascular enhancement of mucinous clot (white arrow, 40 s after contrast infusion). This patient was followed up every 6 months for 15 months and with no signs of malignancy at the end of the follow-up period. (**C**) Typical pancreatic cancer imaging with hypoenhancement and a heterogenous pattern (60 s after contrast infusion). Pathologic diagnosis by surgery: pancreatic ductal adenocarcinoma. (**D**) Atypical pancreatic cancer imaging with a hyperenhancement pattern (90 s after contrast infusion). Pathologic diagnosis by surgery: pancreatic ductal adenocarcinoma. (**E**) Typical pancreatic neuroendocrine neoplasm imaging with an early hyperenhancement (10 s after contrast infusion). Pathologic diagnosis by endoscopic ultrasound-guided fine needle aspiration: pancreatic neuroendocrine tumor, G1 [20]. (**F**) Atypical pancreatic neuroendocrine neoplasm with hypoenhancement and heterogenous pattern (50 s after contrast infusion). Pathologic diagnosis by surgery: pancreatic neuroendocrine tumor, G2 [20]. (**G**) Typical gallbladder carcinoma imaging with an irregular intratumoral vessel (white arrow) and a perfusion defect (white arrowheads) at 120 s after contrast infusion. Pathologic diagnosis by surgery: papillary and tubular adenocarcinoma (pap > tub1). (**H**) Atypical gallbladder carcinoma imaging with a homogeneous enhancement (120 s after contrast infusion). Pathologic diagnosis by surgery: papillary and tubular adenocarcinoma (pap > tub1).

**Table 1 diagnostics-12-01983-t001:** Characteristics of patients who underwent CH-EUS of biliopancreatic lesions (n = 301).

*Patient-Related Information*	
Age, mean (range)	65.8 (23–89)
Number of men	181 (60.1)
Maximum diameter of cyst or mass, mm (SD)	25.3 (±15.1)
Final diagnosis of biliopancreatic lesion	
IPMN ^A^	161 (53.5)
Pancreatic cancer	34 (11.3)
Pancreatic neuroendocrine neoplasm	16 (5.3)
Pancreatic simple cyst	13 (4.3)
Chronic pancreatitis	11 (3.7)
Gallbladder carcinoma	9 (3.0)
Mucinous cystic neoplasm	7 (2.3)
Cholecystitis	7 (2.3)
Autoimmune pancreatitis	6 (2.0)
Gallbladder polyps	6 (2.0)
Serous cystic neoplasm	6 (2.0)
Other malignant diseases ^B^	10 (3.3)
Other benign diseases ^C^	15 (5.0)
Gold standard for final diagnosis	
Clinical follow-up ^D^	192 (63.8)
Surgery	87 (28.9)
Pathology of EUS-FNA	19 (6.3)
Biopsy from metastasis or direct lesion	3 (1.0)
* **Endoscopist- and procedure-related information** *	
Number of diagnoses by dynamic CT	197 (65.5)
Number of diagnoses by MRI	265 (88.0)
EUS trainees (<10 years EUS experience)	10 (71.4)
EUS experts (>10 years EUS experience)	4 (28.6)
Endoscopist certification ^E^	11 (78.6)
Adverse events associated with EUS procedure alone ^F^	5 (1.7)
Iatrogenic Mallory–Weiss tears	2 (0.7)
Gastrointestinal mucosal injury	1 (0.3)
Hypotension during EUS	1 (0.3)
Bradycardia during EUS	1 (0.3)

Unless indicated otherwise, data are presented as n (%). ^A^ Including intraductal papillary mucinous adenoma (n = 129) and intraductal papillary mucinous carcinoma (n = 32). ^B^ Distal bile duct cancer (n = 4), solid pseudopapillary neoplasm (n = 3), pancreatic metastasis of renal cell carcinoma (n = 2), lymphoma of the pancreas (n = 1). ^C^ Gallbladder adenomyomatosis (n = 4), cholelithiasis (n = 3), intrapancreatic accessory spleen (n = 3), healthy normal (n = 2), pancreaticobiliary maljunction (n = 1), acute pancreatitis (n = 1), epidermoid cyst (n = 1). ^D^ Clinical follow-up for at least 6 months when surgical resection is not indicated or when another pathological method could not be performed due to diagnosis of a benign lesion or inoperable malignant disease. ^E^ Board Certification in Japan Gastroenterological Endoscopy Society. ^F^ Side effects of the contrast agent were not observed. CT, computed tomography; MRI, magnetic resonance imaging; EUS-FNA, endoscopic ultrasound-guided fine needle aspiration; CH-EUS, contrast-enhanced harmonic endoscopic ultrasonography.

**Table 2 diagnostics-12-01983-t002:** Diagnostic performances of dynamic CT, MRI, CH-EUS, and all combinations (n = 301).

	Dynamic CT (n = 197)	MRI (n = 265)	CH-EUS (n = 301)	All Combinations ^A^ (n = 301)
Sensitivity (95% CI)	71.3 (61.0–80.1)	70.0 (59.4–79.2)	70.4 (60.8–78.8)	80.6 (71.8–87.5)
Specificity (95% CI)	67.0 (57.0–75.9)	73.1 (65.9–79.6)	78.8 (72.3–84.3)	90.7 (85.7–94.4)
PPV (95% CI)	66.3 (56.3–75.4)	57.3 (47.5–66.7)	65.0 (55.6–73.6)	82.9 (74.3–89.5)
NPV (95% CI)	71.9 (61.8–80.6)	82.6 (75.7–88.2)	82.6 (76.4–87.8)	89.3 (84.1–93.2)

Of the 301 patients, 197 underwent both dynamic CT and CH-EUS, 265 underwent both MRI and CH-EUS, and 161 underwent MRI, dynamic CT, and CH-EUS. ^A^ Dynamic CT, MRI, and CH-EUS. CT, computed tomography; MRI, magnetic resonance imaging; CH-EUS, contrast-enhanced harmonic endoscopic ultrasonography; CI, confidence interval; PPV, positive predictive value; NPV, negative predictive value.

**Table 3 diagnostics-12-01983-t003:** Breakdown of typical and atypical contrast-enhanced harmonic endoscopic ultrasonography imaging in the main diseases where the contrast pattern is considered highly characteristic.

Final Diagnosis	Typical Contrast	Atypical Contrast
IPMA with mucinous clot	110 (85.3) ^A^	19 (14.7)
IPMC with mural nodule	28 (87.5) ^B^	4 (12.5)
Pancreatic cancer	23 (67.7) ^C^	11 (32.4)
Pancreatic neuroendocrine neoplasm	12 (75.0) ^D^	4 (25.0)
Gallbladder carcinoma	22 (68.8) ^E^	10 (31.3)

Unless otherwise indicated, data are presented as n (%). Please note that percentages may not add up to 100% because of rounding or missing values. The representation of the typical contrast in each disease is as follows. ^(A)^ Absence of structures in cyst lumen or pancreatic duct or the presence of noncontrasted structures. ^(B)^ Presence of contrasted structures of >5 mm in the cyst lumen or main pancreatic duct following contrast agent administration. ^(C)^ Presence of heterogenous lesion that was hypodense compared with the adjacent pancreatic parenchyma. ^(D)^ Presence of a homogeneous early enhancement tumor. ^(E)^ Intraluminal mass in the gallbladder, exhibiting irregular intratumoral vessel enhancement or perfusion defect. All cases except ^A–E^ were considered atypical contrasts for each disease. IPMA: intraductal papillary mucinous adenoma; IPMC: intraductal papillary mucinous carcinoma.

**Table 4 diagnostics-12-01983-t004:** Factors in contrast-enhanced harmonic endoscopic ultrasonography that determine correspondence with the final diagnosis using multivariate logistic regression analysis (n = 295).

Independent Variables, n (%)	OR (95% CI)	*p* Value
Age	1.00 (0.98–1.03)	0.84
Men	0.69 (0.37–1.30)	0.26
Other malignant diseases ^A^, 10 (3.3)	Reference	–
IPMN ^B^, 161 (53.5)	6.91 (1.76–27.12)	0.006
Pancreatic cancer, 34 (11.3)	2.88 (0.65–12.69)	0.16
Pancreatic neuroendocrine neoplasm, 16 (5.3)	4.42 (0.81–24.28)	0.09
Pancreatic simple cyst, 13 (4.3)	0.78 (0.14–4.36)	0.78
Chronic pancreatitis, 11 (3.7)	2.63 (0.45–15.44)	0.29
Gallbladder carcinoma, 9 (3.0)	10.52 (0.91–121.90)	0.06
Mucinous cystic neoplasm, 7 (2.3)	2.87 (0.34–24.38)	0.34
Cholecystitis, 7 (2.3)	0.48 (0.06–4.03)	0.50
Autoimmune pancreatitis, 6 (2.0)	Omitted ^C^	NA
Gallbladder polyps, 6 (2.0)	7.79 (0.64–95.14)	0.11
Serous cystic neoplasm, 6 (2.0)	2.59 (0.30–22.18)	0.38
Other benign diseases ^D^, 15 (5.0)	5.23 (0.86–31.83)	0.07

Notably, percentages may not add up to 100% because of rounding or missing values. ^A^ Distal bile duct cancer (n = 4), solid pseudopapillary neoplasm (n = 3), pancreatic metastasis of renal cell carcinoma (n = 2), lymphoma of the pancreas (n = 1). ^B^ Including intraductal papillary mucinous adenoma (n = 129) and intraductal papillary mucinous carcinoma (n = 32). ^C^ Omitted because autoimmune pancreatitis perfectly predicted success. ^D^ Gallbladder adenomyomatosis (n = 4), cholelithiasis (n = 3), intrapancreatic accessory spleen (n = 3), healthy normal (n = 2), pancreaticobiliary maljunction (n = 1), acute pancreatitis (n = 1), epidermoid cyst (n = 1). OR, odds ratio; CI, confidence interval; IPMN, intraductal papillary mucinous neoplasm; NA, not applicable.

## Data Availability

Data can be accessed upon request from the corresponding author.

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
