# Peer review of "Diagnostic Dilemma of Biliopancreatic Contrast-Enhanced Harmonic Endoscopic Ultrasonography"

_diagnostics, 2022, doi:10.3390/diagnostics12081983_

Round 1
Reviewer 1 Report
- Lines 24-25: Consider rephrasing this sentence to enhance clarity.
- Abstract - Lines 28-30: How many patients had IPMN in the present study?
- Introduction: Authors cite 9 references in the first sentence of the section Introduction. They should consider either citing the most relevant and most representative ones, or consider providing more details from all these studies.
- The section Introduction needs to be improved. Namely, it cites a lot of references - 19, but does not provide a lot of information which explain the evidence gap which this study aims to address. For example, consider being slightly more specific about the results of previous studies - e.g. lines 46-47 - why is that so? Lines 50-51 - what does it mean that it remains unclear - were there no previous reports or were results conflicting and if yes how so.
- Line 47: Reconsider the use of the phrase "mainstream diagnostic tool".
- Lines 51-53: This sentence should be clarified - when speaking of diagnostic performance, what was compared with what if the reports mentioned here had only one disease group?
- Lines 58-59: How many were included but did not end up having a confirmed biliopancreatic disease?
- Lines 58-65: Where were these patients recruited at? Which center? Describe very briefly circumstances such as how many patients with biliopancreatic diseases go through this center on an annual basis. What were inclusion and exclusion criteria for patients?
- Line 60: One physician or more? Describe the experience of this physician.
- Line 64: Explain what it means "if necessary" - given that this study aimed to compare CH-EUS and MRI and CT in their diagnostic efficacy in clinical practice.
- Table 1 needs to be improved. Currently it is hard to read, first row is in bold and underlined without any obvious reason. How are EUS trainees given in the Table a characteristic of the patients - as the Table title says. Were the only information collected from the patients who were prospectively included in this study their age and sex? Also, it is unusual to see that in the legend/footnote of Table 1 the reader is told to go to Table 4 to understand what something refers to. This table needs major revisions.
- Lines 92-93: Is this correct - for MRI and CT the experts were board certified, while for CH-EUS there were trainees without board certification who performed the procedure? How does this influence comparisons of the diagnostic utility of the above-mentioned procedures? Would one expect to possibly see an underestimation of the significance of this procedure?
- Line 119: How many patients had all three diagnostic tests? How many two, how many one? What were the criteria for this? How does this affect the statistical analysis?
- Methods: Was sample size calculated and planned?
- Line 141: What does it mean "by the end of the study"? Some became experts during the study period or? Clarify.
- Regarding the references, a third of the cited manuscripts were published a decade or more ago. It would be useful to consider checking whether there might be more recent publications for some of these. For example, the reference No. 5 is a meta-analysis published in 2012. Another group of authors have published a more recent study - please check the following reference: Mei S, Wang M, Sun L. Contrast-Enhanced EUS for Differential Diagnosis of Pancreatic Masses: A Meta-Analysis. Gastroenterol Res Pract. 2019 Mar 6;2019:1670183. doi: 10.1155/2019/1670183. PMID: 30962802; PMCID: PMC6431391.
Author Response
Response to Reviewer 1 Comments
Thank you for the insightful reviewer comments.
We have substantially revised the manuscripts as follows:
Comment 1) Lines 24-25: Consider rephrasing this sentence to enhance clarity.
Response 1) Accordingly, the sentence “To extract the factors (e.g., each disease) that allow the CH-EUS diagnosis to be consistent with the final diagnosis,” was changed to “to extract each disease that allowed the CH-EUS diagnosis to be consistent with the final diagnosis” to enhance clarity at Lines 23‒24.
Comment 2) Abstract - Lines 28-30: How many patients had IPMN in the present study?
Response 2) We thank the Reviewer for this helpful advice. The information regarding the number of IPMN was added at Line 28.
Comment 3) (1) Introduction: Authors cite 9 references in the first sentence of the section Introduction. They should consider either citing the most relevant and most representative ones, or consider providing more details from all these studies.
(2) The section Introduction needs to be improved. Namely, it cites a lot of references - 19, but does not provide a lot of information which explain the evidence gap which this study aims to address. For example, consider being slightly more specific about the results of previous studies - e.g. lines 46-47 - why is that so? Lines 50-51 - what does it mean that it remains unclear - were there no previous reports or were results conflicting and if yes how so.
Response 3) Thank you for the valuable comments.
(1) As per the Reviewer’s suggestions, we have cited the most relevant and most representative references. Hence, we have reduced the number of citations at Line 39.
(2) Accordingly, we have reduced the citations from 19 to 14. Moreover, we made significant revisions to the sentences pointed out by the Reviewer at lines 49‒55 to explain the evidence gap that this study aims to address.
Comment 4) Line 47: Reconsider the use of the phrase "mainstream diagnostic tool".
Response 4) As described above (Response 3), we have added the conditions (i.e., situations requiring pathological diagnosis) under which FNA will be mainstreamed (Lines 45‒47).
Comment 5) Lines 51-53: This sentence should be clarified - when speaking of diagnostic performance, what was compared with what if the reports mentioned here had only one disease group?
Response 5) We have made significant revisions to the sentences at lines 51‒53 to clarify the target of comparison that was not previously reported.
Comment 6) Lines 58-59: How many were included but did not end up having a confirmed biliopancreatic disease?
Response 6) We modified Table 1 by listing the details of the patients who underwent CH-EUS. We also added the sentences “A detailed breakdown of each disease is provided in Table 1” for more clarity in the Method and Result sections at Lines 60 and 159‒160, respectively. Among the patients for whom CH-EUS was requested, none were excluded from this study. Thus, we added the sentence “Finally, of the patients who requested CH-EUS, none were excluded from this study” in the Method section at Line 160.
Comment 7) Lines 58-65: Where were these patients recruited at? Which center? Describe very briefly circumstances such as how many patients with biliopancreatic diseases go through this center on an annual basis. What were inclusion and exclusion criteria for patients?
Response 7) We added the information about “Patient Recruitment Methods,” “Number of biliopancreatic EUS performed,” and “Inclusion and exclusion Criteria” at Lines 60‒70.
Comment 8) Line 60: One physician or more? Describe the experience of this physician.
Response 8) Information regarding the endoscopists who performed CH-EUS has been already listed in the Result section at Lines 163‒164 and Table 1.
Comment 9) Line 64: Explain what it means "if necessary" - given that this study aimed to compare CH-EUS and MRI and CT in their diagnostic efficacy in clinical practice.
Response 9) As mentioned above (Response 7), in the present study, the decision to perform MRI and/or CT taken by the primary physician of each department amd not the endoscopists. Therefore, we added the information “Similarly, at the discretion of each primary physician,” at Lines 71‒72.
Comment 10) Table 1 needs to be improved. Currently it is hard to read, first row is in bold and underlined without any obvious reason. How are EUS trainees given in the Table a characteristic of the patients - as the Table title says. Were the only information collected from the patients who were prospectively included in this study their age and sex? Also, it is unusual to see that in the legend/footnote of Table 1 the reader is told to go to Table 4 to understand what something refers to. This table needs major revisions.
Response 10) Accordingly, we have made significant revisions to Table 1 to illustrate the broad distinction between patient- and procedure-related CH-EUS. We have also ensured that Table 1 is stand-alone and removed the reference comments about Table 4.
Comment 11) Lines 92-93: Is this correct - for MRI and CT the experts were board certified, while for CH-EUS there were trainees without board certification who performed the procedure? How does this influence comparisons of the diagnostic utility of the above-mentioned procedures? Would one expect to possibly see an underestimation of the significance of this procedure?
Response 11) All the diagnoses of CH-EUS, even if a trainee performed the CH-EUS, were made by endoscopists with more than 10 years of EUS experience and a board certification in Japan Gastroenterological Endoscopy Society. Therefore, the misleading sentence was modified at Lines 114‒116. We also added the required information at Lines 105‒106, and Table 1.
Comment 12) Line 119: How many patients had all three diagnostic tests? How many two, how many one? What were the criteria for this? How does this affect the statistical analysis?
Response 12) Of the 301 patients, 197 underwent both DCT and CH-EUS, 265 underwent both MRI and CH-EUS, and 161 underwent MRI, DCT, and CH-EUS. Thus, we added the corresponding information in the Result section at Lines 169‒170, Table 2, and Figure 1. Moreover, we performed “complete case analyses” for these missing values because the missing values occurred at random (MCAR) as previously described (Response 9). Therefore, MCAR allows us to ignore the missing data and still perform a reasonable treatment effect estimation and test, although the number of missing values may reduce the estimation accuracy and statistical power. We added the information in 2.7. Statistical analysis of the Method (Line 153) and Limitations of the Discussion (Lines 304‒307) sections.
Comment 13) Methods: Was sample size calculated and planned?
Response 13) The present study is not an interventional but an observational cohort study. Thus, sample size calculation is not required when complying with the STROBE criteria. Thus, we have added the information that the present study is STROBE compliant in the Method section at Lines 95‒96 to avoid such misunderstandings.
Comment 14) Line 141: What does it mean "by the end of the study"? Some became experts during the study period or? Clarify.
Response 14) The misleading sentence “by the end of the study” has been removed from the manuscript (Line 164).
Comment 15) Regarding the references, a third of the cited manuscripts were published a decade or more ago. It would be useful to consider checking whether there might be more recent publications for some of these. For example, the reference No. 5 is a meta-analysis published in 2012. Another group of authors have published a more recent study - please check the following reference: Mei S, Wang M, Sun L. Contrast-Enhanced EUS for Differential Diagnosis of Pancreatic Masses: A Meta-Analysis. Gastroenterol Res Pract. 2019 Mar 6;2019:1670183. doi: 10.1155/2019/1670183. PMID: 30962802; PMCID: PMC6431391.
Response 15) According to your suggestion, we have replaced the “No. 5” citation with a new citation “Mei S, et, al. 2019” (No.3). Moreover, references with an older publication year were removed in priority order, starting with those that were less relevant to the text in the References section. We were able to reduce the number of old pre-2012 references to one-fifth (Line 335‒411).
In closing, we sincerely thank you once again for your extremely cogent comments that have been instrumental in improving the quality of our manuscript. We hope that the revised manuscript will be deemed suitable for publication.
Sincerely,
Masafumi Chiba M.D., Ph.D.
Department of Endoscopy, The Jikei University School of Medicine
3-25-8, Nishi-Shimbashi, Minato-ku, Tokyo 105-8461, Japan
Tel: +81 3 34331111 ext. 3181; Fax: +81 3 34594524
E-mail: ccl0972009720@gmail.com
Reviewer 2 Report
Dear Dr. Kanazawa and the team,
It was my pleasure in reviewing the above manuscript. I do not see any major concerns.
Best wishes
Author Response
Response to Reviewer 2 Comments
Thank you for the insightful reviewer comments.
We have substantially revised the manuscripts as follows:
Comment) Dear Dr. Kanazawa and the team,
It was my pleasure in reviewing the above manuscript. I do not see any major concerns.
Best wishes
Response) We appreciate your review and positive suggestions for our manuscript.
In closing, we sincerely thank you once again for your extremely cogent comments that have been instrumental in improving the quality of our paper. We hope that the revised manuscript will be deemed suitable for publication.
Sincerely,
Masafumi Chiba M.D., Ph.D.
Department of Endoscopy, The Jikei University School of Medicine
3-25-8, Nishi-Shimbashi, Minato-ku, Tokyo 105-8461, Japan
Tel: +81 3 34331111 ext. 3181; Fax: +81 3 34594524
E-mail: ccl0972009720@gmail.com
Reviewer 3 Report
General comment: The authors presented an interesting original work concerning to the validation of the efficacy of contrast-enhanced harmonic endoscopic ultrasonography (CH-EUS) in clinical practice
The manuscript is written in a comprehensive way.
Title: The title is too long. It should be short, clear, and concise.
Abstract: It is adequate.
The keywords should be different from those used in the title.
Introduction: It is adequate. The authors provided an adequate overview of the thematic.
Methods: The methods are clearly described.
Results: They are presented properly and supported by the Figures and Tables.
Discussion: It is adequate. Tables and Figures should not be cited in the Discussion.
Conclusion: The conclusion is adequate.
References: Adequate.
Recommendation: The manuscript should be accepted for publication in the present form.
Author Response
Response to Reviewer 3 Comments
Thank you for the insightful reviewer comments.
We have substantially revised the manuscripts as follows:
General comment: The authors presented an interesting original work concerning to the validation of the efficacy of contrast-enhanced harmonic endoscopic ultrasonography (CH-EUS) in clinical practice
The manuscript is written in a comprehensive way.
Comment 1) Title: The title is too long. It should be short, clear, and concise.
Abstract: It is adequate.
Response 1) We agree with you, and we changed the title from “Application Potential of Biliopancreatic Contrast-Enhanced Harmonic Endoscopic Ultrasonography in Clinical Practice: A Single-Center Prospective Study” to “Diagnostic Dilemma of Biliopancreatic Contrast-Enhanced Harmonic Endoscopic Ultrasonography” to be short, clear, and concise (Lines 2‒3).
Comment 2) The keywords should be different from those used in the title.
Response 2) Informative keywords that do not overlap with the title were extracted using the MeSH function. Thus, we changed “contrast-enhanced harmonic endoscopic ultrasonography (CH-EUS); diagnostic performance; biliopancreatic lesion” to “ultrasound contrast agents; endosonography; diagnostic performance; biliary tract diseases; pancreatic diseases” (Lines 33‒34).
Introduction: It is adequate. The authors provided an adequate overview of the thematic.
Methods: The methods are clearly described.
Results: They are presented properly and supported by the Figures and Tables.
Comment 3) Discussion: It is adequate. Tables and Figures should not be cited in the Discussion.
Response 3) All table and figure citations have been removed from the Discussion section (Lines 263‒309).
Conclusion: The conclusion is adequate.
References: Adequate.
Recommendation: The manuscript should be accepted for publication in the present form.
General Response) T Thank you for thoroughly perusing our manuscript and pointing this out.
In closing, we sincerely thank you once again for your extremely cogent comments that have been instrumental in improving the quality of our paper. We hope that the revised manuscript will be deemed suitable for publication.
Sincerely,
Masafumi Chiba M.D., Ph.D.
Department of Endoscopy, The Jikei University School of Medicine
3-25-8, Nishi-Shimbashi, Minato-ku, Tokyo 105-8461, Japan
Tel: +81 3 34331111 ext. 3181; Fax: +81 3 34594524
E-mail: ccl0972009720@gmail.com
Round 2
Reviewer 1 Report
I would like to thank the Authors for revising their paper according to the provided remarks.
Comments: Introduction - Reconsider once again the use of term "mainstream". It seems more appropriate to find a similar phrase which would indicate a tool which is used the most and/or used more and more, and which would correspond more with academic writing. Regarding Authors' answer for sample size - the answer is not appropriate. Observational studies too, not only interventional studies, also have methods for deciding on sample size. And, adding that the study is compliant with STROBE guidelines (did you also add the checklist?) does not mean that no plan was necessary when it comes to sample size. Now that you have stated that the study follows STROBE guidelines - please add the STROBE checklist which includes the question regarding how the sample size was arrived at. The text which Authors have added needs to be checked for English language grammar.Author Response
Response to Reviewer 1 Comments (R2)
Thank you for the insightful reviewer comments.
We have substantially revised the manuscripts as follows:
Comment 1)
(1) Introduction –Reconsider once again the use of term "mainstream". It seems more appropriate to find a similar phrase which would indicate a tool which is used the most and/or used more and more, and which would correspond more with academic writing.
(2) Regarding Authors' answer for sample size - the answer is not appropriate. Observational studies too, not only interventional studies, also have methods for deciding on sample size. And, adding that the study is compliant with STROBE guidelines (did you also add the checklist?) does not mean that no plan was necessary when it comes to sample size. Now that you have stated that the study follows STROBE guidelines - please add the STROBE checklist which includes the question regarding how the sample size was arrived at.
(3) The text which Authors have added needs to be checked for English language grammar.
Response) Thank you for your advice.
(1) We changed “mainstream tool” to “typical tool” at Line 46.
(2) The sample size was calculated as follows: Assuming sensitivity of 70% in the CH-EUS group and 80% in all combination group, with a type I error of 0.05 (two-sided) and a power of 0.8, a minimum of 294 patients in each group were required. Thus, we added the sample size information in the Method section at Lines 153‒155. We also attached the STROBE checklist as a supplementary file.
(3) The previously added text has been carefully reviewed by an experienced editor whose first language is English (Enago editing service). Therefore, we attached the English proofreading certificate for the added text.
We appreciate your review and positive suggestions for our manuscript.
Sincerely,
Masafumi Chiba M.D., Ph.D.
Department of Endoscopy, The Jikei University School of Medicine
3-25-8, Nishi-Shimbashi, Minato-ku, Tokyo 105-8461, Japan
Tel: +81 3 34331111 ext. 3181; Fax: +81 3 34594524
E-mail: ccl0972009720@gmail.com